# Influence of Moisture in Concrete on the Photothermal Response—A New Approach for a Measurement Method?

**Rainer Krankenhagen**

Bundesanstalt für Materialforschung und -Prüfung (BAM), Unter den Eichen 87, 12205 Berlin, Germany; rainer.krankenhagen@bam.de

**Abstract:** Recently, the photothermal determination of the thermophysical properties of concrete under lab conditions was reported. Their values are mainly needed to look at the energy consumption of buildings. Additionally, changes in their values in relation to the initial state might also be a good indicator for material quality or for moisture. The present contribution explains the photothermal method in a more general way to indicate the potential for on-site application. Secondly, a special application case is regarded: the detection of moisture in concrete. Two concrete samples were soaked with water, followed by a drying period, to obtain different levels of water penetration. The water contents were determined by weighing, and the photothermal response was measured. The results show a large influence on the measured temperature transients, which is larger than expected from the original simple model. They clearly provide two points: the photothermal method is suited to detect moisture in concrete, but the magnitude of the actual measurement effect is not yet understood.

**Keywords:** thermal effusivity; thermal conductivity; moisture; reflectivity

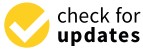



## 1. Introduction

The use of infrared cameras for measuring surface temperature distribution for the evaluation of buildings and structures or parts of them is common practice in civil engineering. The so-called passive approach is popularly used, whereby only one infrared image of the inspected surface under certain environmental conditions is recorded and evaluated. A series of standards describing the procedure for different application cases such as for buildings is available, either generally [1–4] or for bridge decks [5]. On the contrary, the active approach comprises an intended heating or cooling of the inspected surface and the recording of the surface temperature transient. This procedure requires relatively more equipment, time, and experience for evaluation. However, it can provide more reproducible and reliable information about the inspected component as compared to the passive approach. However, its application in civil engineering is limited, and most of the published results are in research and development work. A standard is not yet available for the field of civil engineering. The vast majority of published reports focus on inner detachments with inner structural features or with inner defects in general.

On the other hand, for components with well-known or very large thicknesses, the active approach can be exploited to determine thermophysical properties of the respective material. Thus, active thermography can be used for both, i.e., non-destructive failure detection, as well as material characterization. Furthermore, if the material state or composition differs from its regular state, as in the case of moisture, this could also be detectable. In 2007, Poblete and Pascual reported the influence of moisture in a cement matrix on the thermal diffusivity of the material and related the results to ultrasound velocities [6]. A similar approach was recently used by Mayr et al. to monitor the degree of curing in a polymer from a thermal response [7].

Moisture in building materials is a widespread issue that influences the thermophysical properties. Moisture influences not only the pure material properties but also surface

temperature and appearance. The surface temperature can be reduced in relation to the ambient temperature due to evaporative cooling [8]. However, the interaction between water transport and temperature is quite complex and results in a specific surface temperature transient with three different phases [9]. Thus, infrared cameras are often used to study the dynamics of drying processes in building materials (see, for example, [10–14]), especially in order to determine the moisture retention curve for the respective material [15]. The German standard DIN EN 16682 [16] deals with moisture in cultural heritage and mentions thermography in its appendix (C.8) under "further relative methods" as a qualitative method.

Recently, a robust measurement method was reported that is able to characterize the thermal behavior of concrete under lab conditions and is convertible into an on-site method [17]. The surface to be investigated is heated shortly by a laser pulse, and the cooling down is recorded by an infrared camera without contact. This photothermal method was applied to three different concrete types and was found to work well on dry material. Since moisture influences the thermal material properties of heat conductivity and heat capacity [18,19] the photothermal method should be sensitive to moisture. However, it remains unclear whether the expected effect is detectable by the available technical equipment and whether the effect can be described within the framework of existing models.

The idea of this study is to investigate the influence of moisture on the measurement results of the photothermal method. Here, the same concrete samples as those used in [17] were soaked in different cycles to obtain different moisture contents and investigated with the same equipment as in [17]. After a short presentation of the photothermal method (concept and setup) and the results obtained with dry concrete, the soaking procedure is reported. Based on the present theory and the obtained moisture contents, the expected effect size is considered. The results of moist concrete are then presented and discussed. The expected effects were observed, but the magnitude of the effect was larger than previously estimated and could not be understood in terms of the simple model successfully applied in the case of dry concrete.

## 2. Concept of the Photothermal Approach

During and after heating of the surface of a solid, a temperature gradient arises between the surface and the bulk, leading to a heat flow into the volume of the solid. This heat flow can be described by the well-known heat equation provided by numerous textbooks (see, for example, [20,21]). Since it is a partial differential equation of the second grade, its solutions are mainly determined by spatial and temporal boundary conditions. However, for certain conditions and simplifications, analytical solutions have been found and published in textbooks [20,21]. For the intended application, i.e., the determination of thermal properties in an on-site measurement, a relatively simple equation can be derived:

$$\Delta T(t) = \frac{Q}{\sqrt{\pi} \cdot \varepsilon \cdot \sqrt{t}} \tag{1}$$

Equation (1) describes the surface temperature decrease after heating the surface with a Dirac pulse with the absorbed spatial energy density ($Q$, given in J/m$^2$) at t = 0. Here, $\Delta T(t)$ stands for the temperature change in relation to the initial surface temperature (T0) before the pulse. The assumed simplifications are:

1. The material properties and surface heating are completely isotropic;
2. The initial temperature distribution is isotropic and equal to the ambient temperature;
3. Theoretically, the material must be semi-infinitely thick, but in reality, it is sufficient for the temperature on the rear side to remain unchanged during the observation period;
4. The entire surface is heated, and edges are not considered, which means that heat flows only in one direction, i.e., perpendicular to the surface; this is a one-dimensional consideration of the depth profile of the temperature;
5. The material is opaque, which means that the entire heat energy is absorbed within an infinitesimally thin surface layer;

6.  Thermal heat losses due to convection and irradiation are neglected.

The thermal properties of the material are included in $\varepsilon$, which is the thermal effusivity formed by the simple product of density ($\rho$), thermal conductivity ($k$), and specific heat capacity ($c$):

$$\varepsilon = \sqrt{k\rho c} \tag{2}$$

Thus, any property of a material that affects one of these material parameters will cause a change in the thermal effusivity. In the case of mixtures of materials, the effusivity of the mixture is affected by the mixture ratio of the respective components (if they have different material properties, which is usually the aim of a mixture).

Coming back to a concept for an on-site inspection: Equation (1) can be used to determine the actual material effusivity ($\varepsilon$) when $Q$ is known and $T$(t) is correctly recorded. The implementation in practice faces some challenges, which are compiled in Table 1, together with their solutions.

**Table 1.** Overview of challenges and possible solutions for a technical implementation of the photothermal approach as an on-site inspection method.

| Challenge | Solution |
| --- | --- |
| Heating the surface with a Dirac pulse | Photothermal heating by a short pulse from a flash lamp or a laser with beam shaping while the pulse length should be much shorter than the observation period |
| Measuring the actual absorbed energy ($Q$) | Knowing the reproducible energy density released from the used external energy source ($Q_{ext}$), the absorbed energy density ($Q$) can be calculated by subtracting the actual reflected energy on the inspected surface using the reflection coefficient ($R$): $$Q = Q_{ext}(1 - R)$$ where $R$ is an effective parameter and results from the integral of the spectral intensity distribution of the source weighted with the reflection spectrum of the surface |
| Measuring the fast temperature decrease | Using an infrared camera with a sufficiently high frame rate ($> \approx 10$ Hz) or a fast pyrometer with emissivity correction |
| Selection of a suitable zero point for the time scale in relation to the pulse length | Taking the half of the pulse length is mathematically correct (Dirac pulse is the limit of a series development of square pulses with area $Q$ of decreasing pulse width) |

Thus, a technical implementation of the photothermal approach for concrete surfaces requires only three steps:

1.  Well-defined heating of a certain area with a short pulse ($\leq 1$ s);
2.  Measurement of the actual reflection coefficient in this area for the applied pulse; and
3.  Measurement of the temperature decrease for about 1 min after the pulse with at least 0.1 s temporal resolution.

The basic setup is illustrated in Figure 1.

The final determination of the effusivity value can be achieved by a non-linear regression of Equation (1) or, even more simply, by calculating the value at a certain time. A successful application of this simple model was also demonstrated earlier to determine the absorbed energy of flash lamps [22]. In addition, considering the curve shape of the temperature transient allows for verification of whether the use of Equation (1) is useful for the recorded measurement dataset.

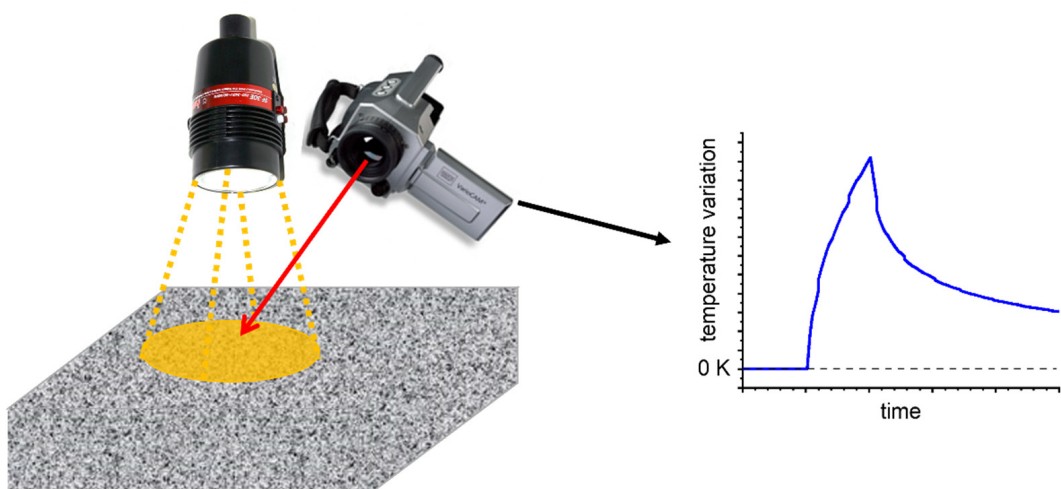

**Figure 1.** Schematic of the technical implementation of the photothermal method. The red arrow indicates the optical axis of the infrared camera; the diagram with the blue curve represents the measured surface temperature transient.

## 3. Experimental Setup

The setup shown in Figure 1 was realized under laboratory conditions. Figure 2 illustrates the experimental setup; it is rotated by 90° in comparison to the scheme.

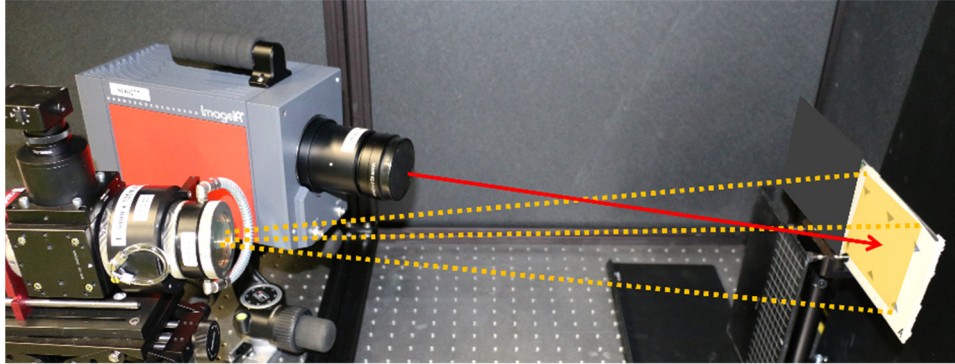

**Figure 2.** Experimental setup under lab conditions. The orange dotted lines indicate the beam shape of the laser optics on the left side; the infrared camera is located behind. With the related optical axis, the inspected object can be seen on the right margin, where the orange rectangle indicates the heated region ($13 \times 13$ cm$^2$ area) of the object.

The use of a laser with beam-shaping optics has some advantages in comparison to traditional flash lamps or halogen lamps. It enables spatially homogeneous heating with a well-defined pulse shape without afterglow or intensity variations during the pulse. Figure 3 demonstrates the spatial alignment of the sample and heated area. The technical parameters of the laser system are:

- Output power at the fiber: 540 W;
- Wavelength: 935–940 nm (depending on output power);
- Pulse length: 10 ms to infinity;
- Beam width after optics: $130 \times 130$ mm$^2$ square at 60 cm distance;
- Irradiance measured at this distance: $3.0 \pm 0.1$ W/cm$^2$.

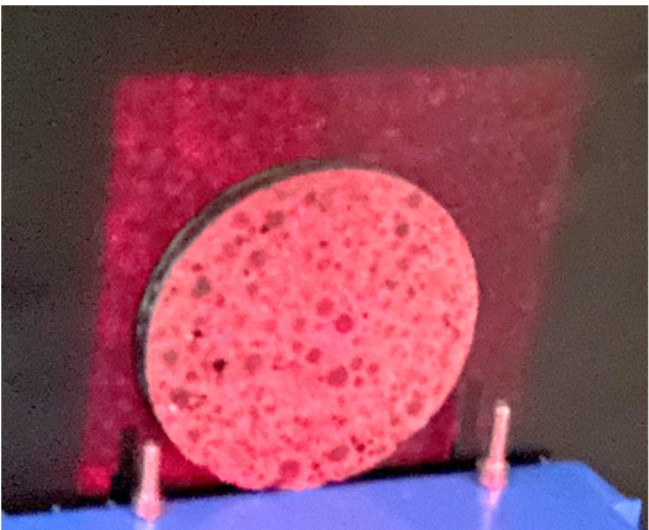

**Figure 3.** Photo of the heated area in relation to a circular specimen with 10 cm diameter (the area is illuminated by a red pilot laser).

The technical specifications of the used infrared camera, an Image IR 8300 with a 25 mm objective, are:

- Cooled InSb detector array with $640 \times 512$ pixels;
- Frame rate: 300 Hz full frame;
- Sensitivity range: 2.5–5.7 μm;
- NETD: <25 mK;
- Integration time (10–100 °C): 0.5 ms.

The settings for the experiments were:

- Heating pulse length: 0.5 s;
- Observation period: 1–2 min variable;
- Frame rate: 40 Hz;
- Distance between camera and sample surface: 61 cm;
- Resulting spatial resolution: 5 pixels/mm.

The measurements of the hemispherical reflectivity at 940 nm were conducted in a separate experiment with a Perkin-Elmer Lambda 1050s spectrometer on different parts of the inspected specimen to cover the different additives, as well as the cement matrix present on the surface. One measurement covered a circular area with a 2.5 cm diameter. More details can be found in [17].

## 4. Concrete Samples

### 4.1. Dry Samples

For the quantitative evaluation method, three different types of concrete were selected: one repair mortar and two lightweight concretes with different clays. The different materials are shown in Figure 4. A large part of the surface was coated for other investigations. As described in Section 2, a coating influences the photothermal signal due to a change in reflectivity and must be considered during evaluation.

The main material parameters of the different concretes are compiled in Table 2. Thermal conductivities were measured using a commercial device based on the TPS (transient plane source) method, details of which are reported in [17].

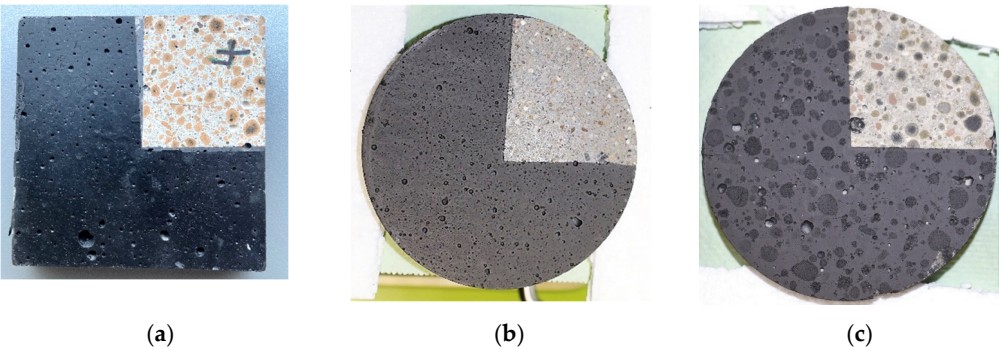

(**a**)                                    (**b**)                                    (**c**)

**Figure 4.** (**a**) Partially coated concrete sample type 1 (side length: 10 cm); (**b**) partially coated concrete sample type 2 (diameter: 10 cm); (**c**) partially coated concrete sample type 3 (diameter: 10 cm).

**Table 2.** Density and thermal conductivity of the investigated concrete samples.

| Property | Type 1 (Lightweight Concr. Red Expanded Clay) | Type 2 (Repair Mortar) | Type 3 (Lightweight Concr. Black Expanded Clay) |
|---|---|---|---|
| Density (g/cm$^3$) | 1930 | 2125 | 1800 |
| Thermal conductivity from TPS (W/(m · K)) | 1.37 | 2.54 | 1.6 |
| Thermal conductivity according to the German standard DIN 4108-4 (W/(m · K)) | 1.35 | 1.6 | 1.15 |

Table 2 also provides values for thermal conductivity according to the German standard DIN 4108-4 [23], where a direct correlation between density and thermal conductivity is reported. The measured values of two samples (types 2 and 3) obtained by TPS are relatively high (by a factor of 1.5) in relation to the requirements of the standard. This result cannot be discussed here but is an indication of the need for on-site measurements.

*4.2. Moist Samples*

Similar to Poblete and Pascal, different moisture contents were obtained by simply soaking the concrete samples in water and estimating the moisture content by weighing [6]. However, the existing samples were partially coated (see above), and the coating reduces the water absorption by covering and closing micropores on the surface. Thus, the coatings had to be removed. Figure 5 shows the appearance of the prepared samples after the coating was removed. Here, only type 1 and type 3 were further investigated. The type 1 sample is shown from both sides, which have different appearances. Regarding an on-site application, the formwork side (Figure 5b) represents the more realistic case. The type 3 sample was cut as a slice from a cylinder; thus, there is no formwork side on this sample. Please note that certain pores seem to contain residues from the coating; thus, the water penetration might be influenced in comparison to the original state of the concrete.

Both samples were soaked in pure water at room temperature. Starting with a long-term soaking of 19 days, further cycles of drying and soaking were carried out to obtain different water contents. However, the water content was sometimes not stable within a few hours; thus, a series of mass measurements was necessary, and gaps between measurements were interpolated. Figure 6 shows the measurement results for the first soaking experiment.

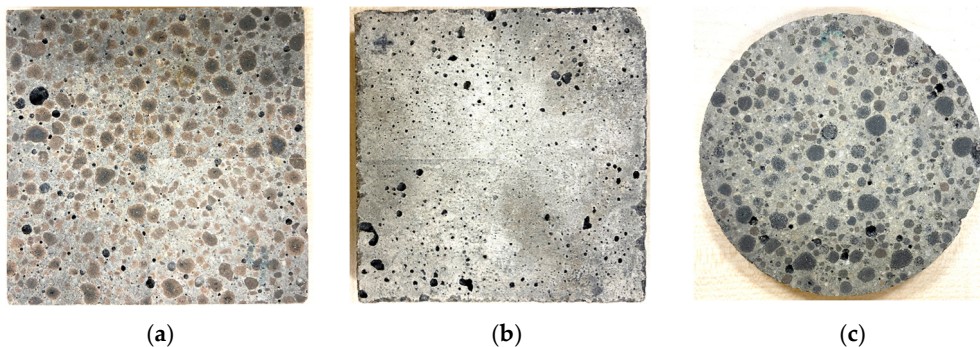

**Figure 5.** (**a**) Concrete sample type 1, cut side (after removing the coating); (**b**) concrete sample type 1, formwork side (after removing the coating); (**c**) concrete sample type 3 (after removing the coating).

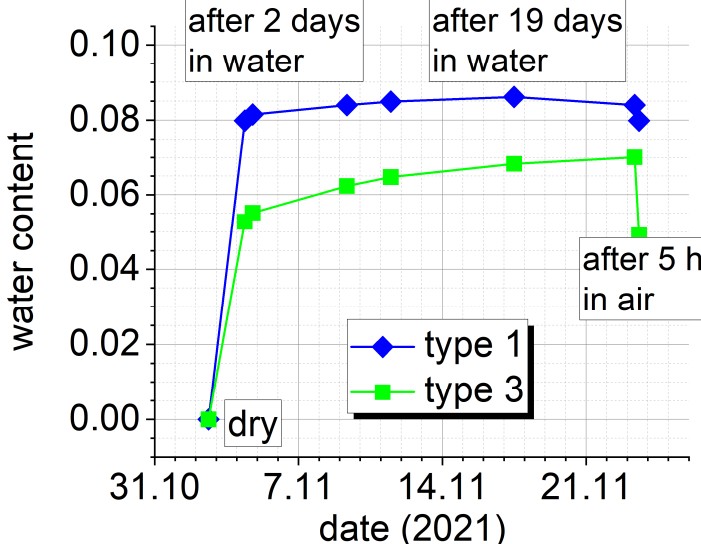

**Figure 6.** Increase in water content (moisture) during a period of 19 days of soaking and a sudden drop after 5 h drying in air.

Most of the moisture was absorbed within the first two days. The two materials exhibited different absorption dynamics, although they are very similar (appearance and material properties reported in Table 2). The difference is even more pronounced during drying; the type 3 sample lost around one-third of the water content within 5 h.

Figure 7 illustrates the chronology of several soaking/drying cycles. Please note the different starting levels in spring 2022 due to different sample storage conditions. Here, the drying of the sample type 1 sample began 21 days earlier. The numbers of days in the boxes refer to the type 3 sample. Table 3 lists the treatment data again to provide a better overview of the samples' histories. It is clear that the estimated water content is an average over the whole volume. Due to drying out through the surface, lower values for moisture content would be expected in the near-surface region. However, the averaged values are used as an indicator of moisture content.

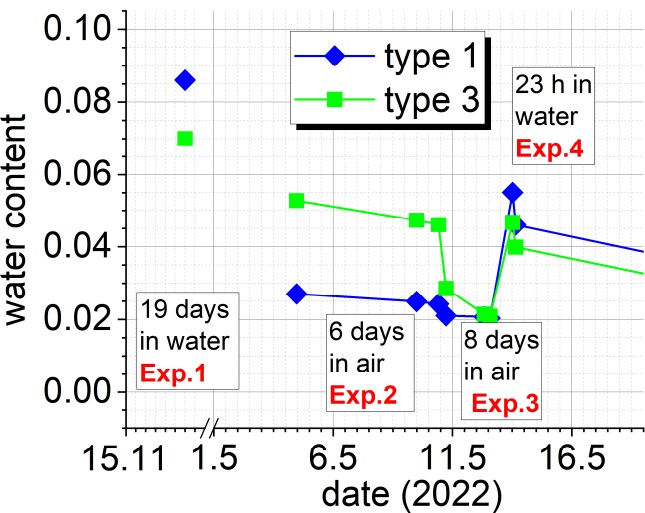

**Figure 7.** Variation of water content during several soaking/drying cycles. Red font indicates the corresponding photothermal measurements; day data in the boxes refer to type 3.

**Table 3.** Chronology of soaking/drying cycles and the related photothermal investigations.

| Date | State of Type 1 (Lightweight Concr. Red Expanded Clay) | State of Type 3 (Lightweight Concr. Black Expanded Clay) | Photothermal Investigation |
|---|---|---|---|
| 21 November 2021 | Saturated after 19 days in water | Almost saturated after 19 days in water | Exp. 1 |
| 21 November 2021 | 5 h drying in air | 5 h drying in air | No exp. |
| 5 February 2022 | Soaking | Soaking | No exp. |
| 13 April 2022 | Drying in air | Covered with foil | No exp. |
| 4 May 2022 | In air | Drying in air | No exp. |
| 10 May 2022 | After 27 days drying | After 6 days drying | Exp. 2 |
| 10 May 2022 | 5 h in sun | 5 h in sun | No exp. |
| 12 May 2022 | After 1 month drying | After 8 days drying | Exp. 3 |
| 13 May 2022 | Partially saturated after 20 h soaking | Partially saturated after 20 h soaking | Exp. 4 |
| 7 June 2022 | Drying in air | Drying in air | No exp. |

### 4.3. Preliminary Consideration of the Photothermal Detectability

In order to estimate the influence of moisture on effusivity, the correlation between moisture and the separate physical properties has to be evaluated. The final values were calculated with the following assumptions:

1. The density rises linearly with the sample mass (the volume remains constant);
2. The heat capacity is the sum of capacities of concrete and water weighted by their respective mass parts (water in the micropores adds additional heat capacity to the heat capacity of dry concrete); and
3. The thermal conductivity rises linearly with increasing water content, with the slope depending on the specific concrete microstructure. A slope value of 4 was selected relative to the mass percentage of water in the concrete.

It is important to note that different relationships between thermal conductivity and moisture were reported in the past. Steier and Hurd noted a slope value of 5% for concrete as a general rule [24]. Kodide reported a series of measurements in different kinds of

concrete and found values between 4% and 8%, with values up to 15% for specific ternary mixtures [25]. The German IBP published several values for several materials in a table including expanded clay concrete [26]; thus, a value of 4% was selected here. In the case of the type 1 sample, a maximum mass increase of 8.8% was reached after soaking. With the aforementioned assumptions, the effusivity of the wet concrete would increase by 37% (from 1630 to 2200 $W \cdot \sqrt{s}/\text{K} \cdot \text{m}^2$). As shown later at the end of Section 5.1, this difference should be easily detectable with the photothermal method. If the thermal diffusivity with the same material parameters is considered, a reduction of only 5% would be expected (from 7.1 to 6.84, both in $10^{-3}$ cm$^2$/s).

However, this is only a preliminary consideration based on Equation (1) with the six mentioned simplifications, which was successfully applied in the case of dry concrete. In particular, the assumptions of isotropic material properties and a homogeneous temperature distribution within the material are probably not fulfilled, taking into account moisture and the resulting effusivity gradients within the material and the convective cooling taking place at the surface. However, it is unknown how strongly these effects influence the photothermal response.

## 5. Experimental Results

### 5.1. Results from Dry Concrete

In the following part, thermograms only from the type 1 sample are presented exemplarily; more data were previously reported in [17]. Figure 8 shows the very first thermogram after the pulse had started (within the 25 ms preceding period). It is displayed as difference thermogram, i.e., the initial temperature distribution before the pulse was subtracted to obtain a temperature difference as contained in Equation (1). The thermogram clearly shows the difference between the coated and the uncoated part (the upper-right quarter is uncoated). Investigating both cases is important for a later on-site application, where different kinds of coatings may occur. Figure 9 shows the thermogram after 20 s; the surface temperatures decreased, but a difference between the coated and the uncoated sections remains distinctly visible.

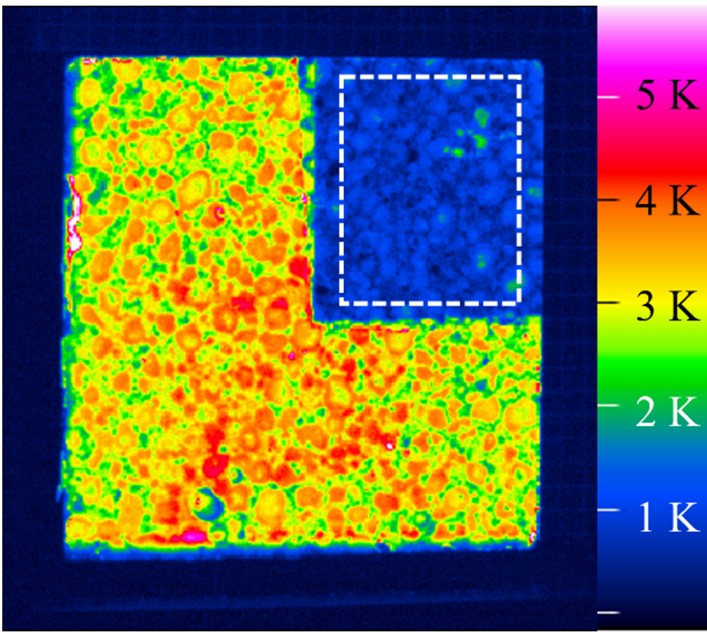

**Figure 8.** First difference thermogram of the type 1 sample after the pulse had started (heating duration < 25 ms). The white dashed rectangle includes the measurement region for the temperature transients.

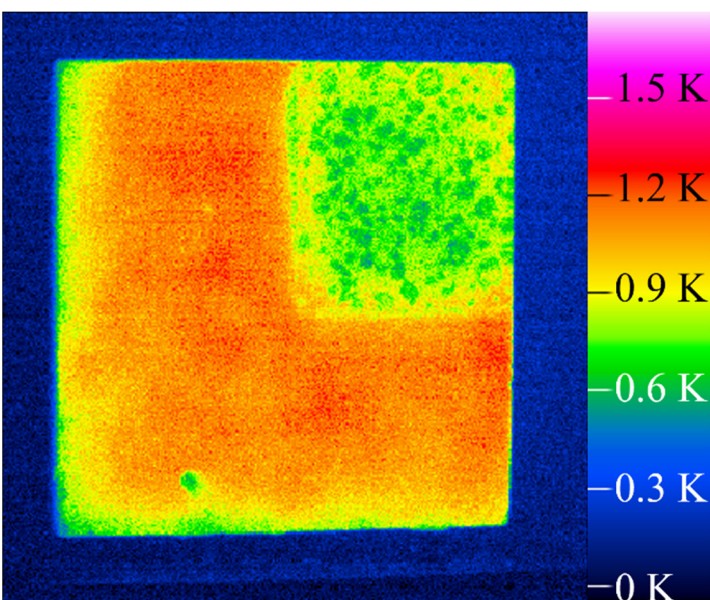

**Figure 9.** Difference thermogram of the type 1 sample after 20 s cooling down.

For the determination of temperature transients, the part of the surface enclosed by the rectangle in Figure 8 within the uncoated region was selected to achieve an averaging over the grains. The obtained curve is displayed in Figure 10. As introduced in the concept section (Section 2), the 0.5 s pulse is very short in comparison to the observation period. The obtained transient shows exactly the expected behavior from Equation (1): the surface temperature curve forms a line in the double log plot after 0.6 s (Figure 11). After 10 s, the temperature decrease slows down as a result of the small sample thickness of 1 cm. Here, the third simplification mentioned in the concept section (the material is semi-infinitely thick) is no longer valid. Therefore, evaluation of the effusivity according to Equation (1) should be performed within a certain period (in this example, 10 s).

A comparison of all obtained effusivity values is presented in Figures 12 and 13, together with the requirement of the standard (see Table 2). Details of the uncertainty limits were reported in [17].

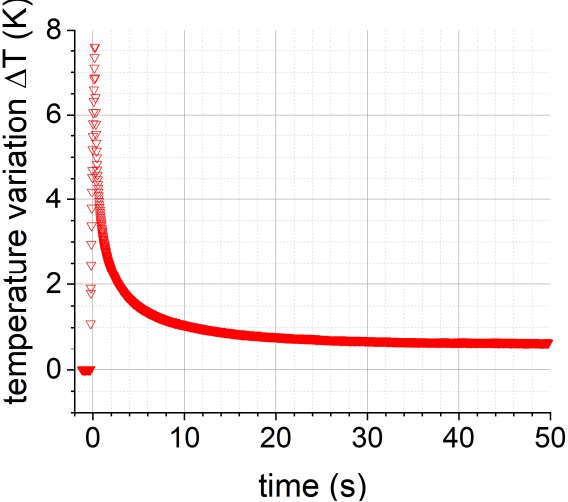

**Figure 10.** Averaged surface temperature transient recorded for the type 1 sample during and after the 0.5 s laser pulse (averaged over the white dashed rectangle in Figure 8).

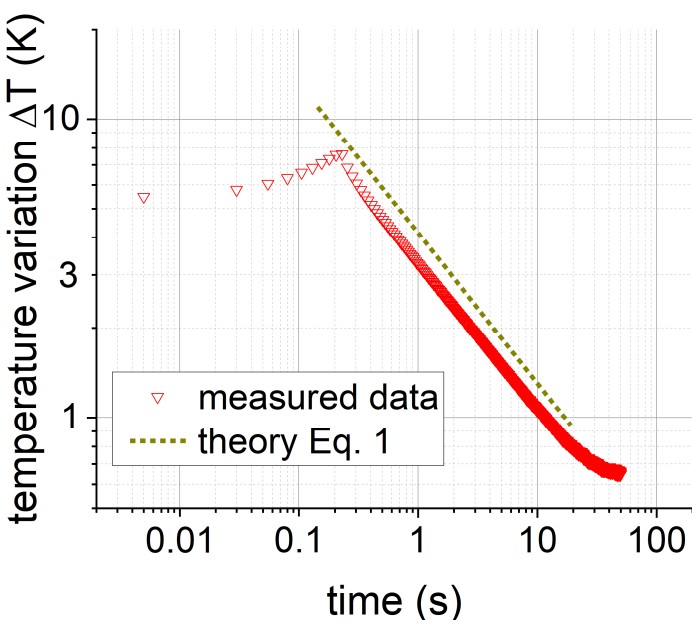

**Figure 11.** Same dataset same dataset as in Figure 10 but as a log–log plot to enable a comparison with Equation (1).

Figure 12 demonstrates that the effusivity can be determined on both uncoated and coated surfaces. Additionally, type 2 sample differs from the requirement of the standard, as also shown in Table 2. It is important to note that this material was classified as repair mortar; therefore, it is technically not concrete and differs from the other two materials in terms of the value of effusivity. However, type 1 and type 3 cannot be distinguished within the given uncertainties.

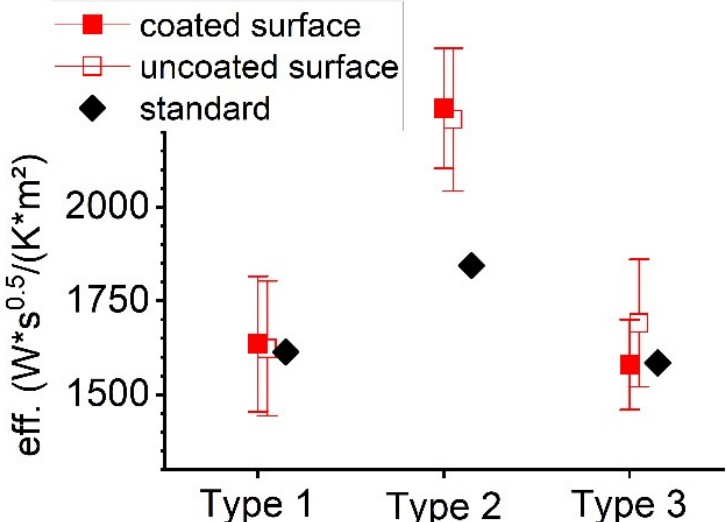

**Figure 12.** Overview about the effusivities of the three investigated concrete samples with their uncertainties and the value required by the standard [23].

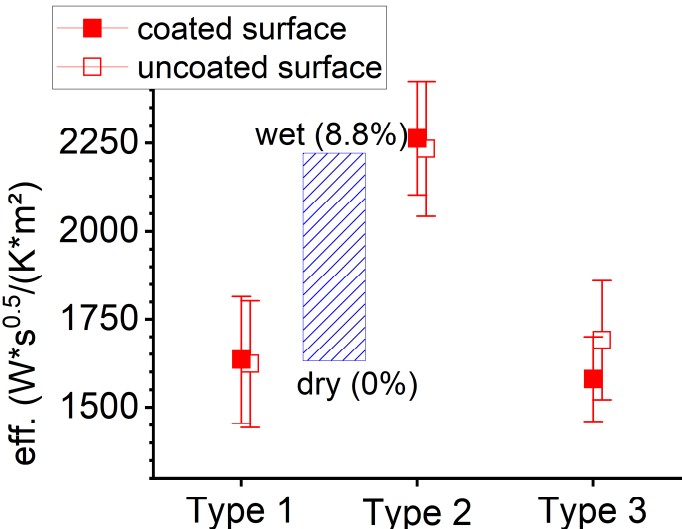

**Figure 13.** Measured effusivities with their uncertainties (same data as in Figure 12) in relation to the estimated effect size of moisture influence, as indicated by the blue striped rectangle (expected dynamic range).

It is important to highlight that the reported uncertainties of the photothermal method were estimated from first investigations; thus, there is potential for further improvements. The possibility of detecting effusivity differences is the basis of the proposed approach. In Figure 13, the estimated influence of moisture on the effusivity (numbers from Section 4.3) is displayed in relation to measurement values and their uncertainties (the blue striped rectangle in comparison to the error bars). One can conclude that saturated moisture detection should be relatively straightforward, and the detection of partially dried material might be possible. Thus, it is interesting to study the experimental results obtained for material(s) with different water contents.

### 5.2. Results from Moist Concrete

The thermographic observation of a moist surface reveals a striking difference relative to the case of a dry concrete surface, which is demonstrated by Figures 14 and 15.

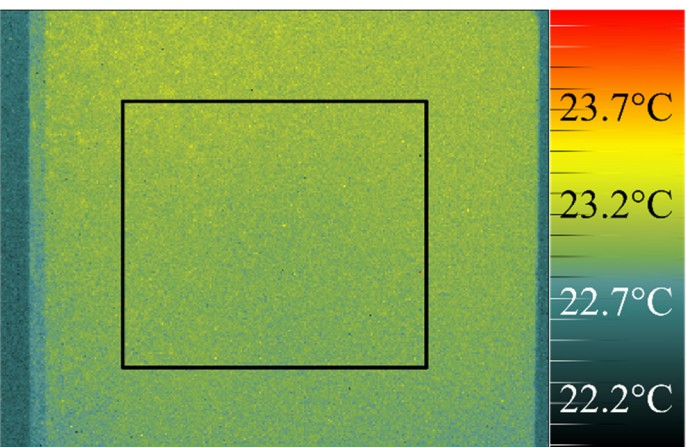

**Figure 14.** Thermogram of the dry type 1 sample before pulse heating. The black rectangle indicates the region used for extraction of the transient.

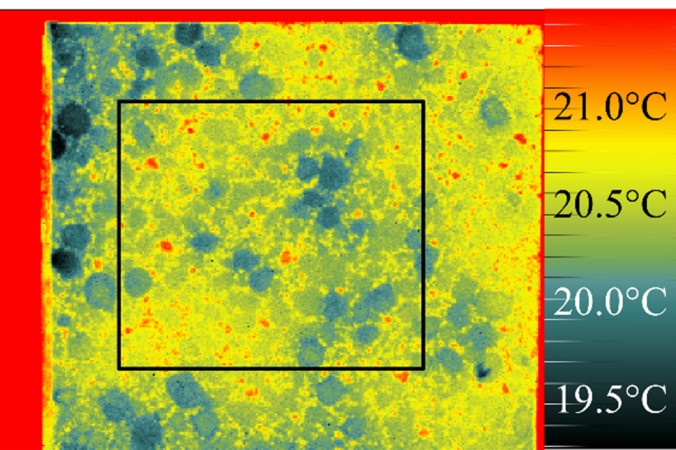

**Figure 15.** Thermogram of the wet (5.4% water content) type 1 sample before pulse heating. The black rectangle indicates the region used for extraction of the transient.

The initial temperature distribution on the surface of the type 1 sample is displayed in the dry state (Figure 14), as well as in the moist state with 5.4% water content (Figure 15). To enable a useful comparison, the displayed temperature ranges have the same dynamics but are shifted by 2.7 K with respect to each other. The background in both cases has the same temperature of 22.5 °C. In the case of a moist surface, the temperature is certainly lowered, and features of the grain structure are recognizable without a heat pulse. This is a result of evaporative cooling taking place at the surface of the wet concrete independent of the respective drying phase [9,14]. In the shown case, the cut clay grains appear colder, probably due to their porous consistency with enhanced capillarity for water transport, leading to an enhanced evaporation.

As already explained in Section 2, the reflection coefficient of the investigated surface must be known to correctly determine the absorbed pulse energy. As known from common practice, moist concrete appears darker than when dry. Parrott pointed out the possibility of detecting moisture by optical reflection [19], and Ludwig reported the same effect for pine [18]. Therefore, the reflection coefficient must be known for every specific moisture content in order to apply Equation (1) with the correct value of $Q$.

Figure 16 presents measured reflectivity values of the investigated samples. There is a general trend of a decrease in reflectivity with increasing moisture content. However, the measurement itself is not very accurately indicated by the error bars (determined from 4–5 single measurements at different positions). The respective percentage of the darker inclusions within the analyzed region (2.5 cm diameter) influences the obtained value.

The temperature transients for the type 1 sample on the cut side are shown in the following figures as previously shown in Figures 10 and 11, i.e., both include the same datasets but with different axis scales.

Both plots contain six datasets of six separate measurements. The transient for 0% is not comparable with the others because the surface was completely free of coating material (see explanation in Section 4.2). Thus, the related curve demonstrates only the general shape and a trend. The obtained transients presented in Figure 17 reveal the expected behavior in principle: the highest moisture content is related to the lowest temperature transient due to the enhanced effusivity. With decreasing moisture content, the surface temperatures during and after the pulse reach higher values. Please note that this is not a simple shift of the curve but a stretching of the curve shape. However, the small difference between Exp. 2 and Exp. 3 (2.4% and 2.0% moisture, respectively) could not be resolved, while the difference between 5.5% and 5.2% moisture was detectable. Figure 18 displays the transients in a log–log plot to compare the curve shape with the prediction of Equation (1), shown as dashed line with an arbitrary intercept. Additionally, the blue rectangle indicates the expected effect size (vertical extent) for a hypothetical concrete, similar to Figure 13.

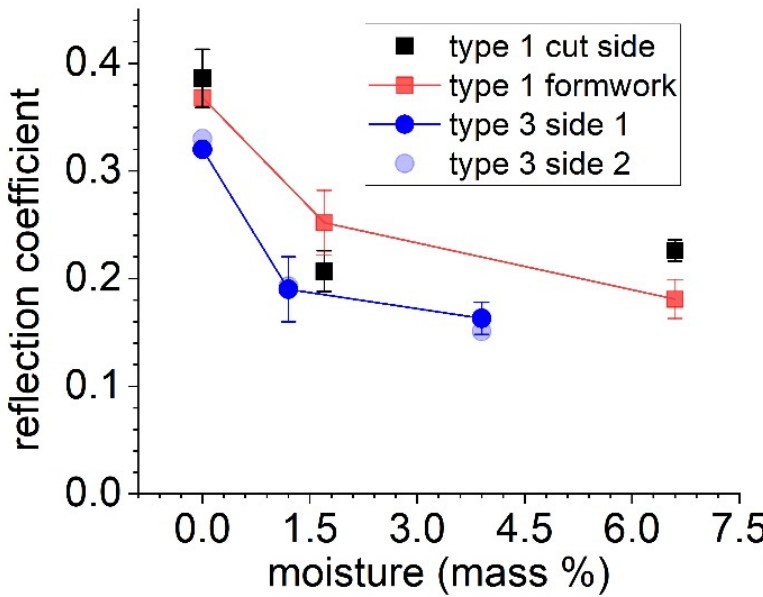

**Figure 16.** Reflection coefficient at 940 nm measured on sample surfaces with different moisture contents. The error bars result from multiple measurements in different surface regions.

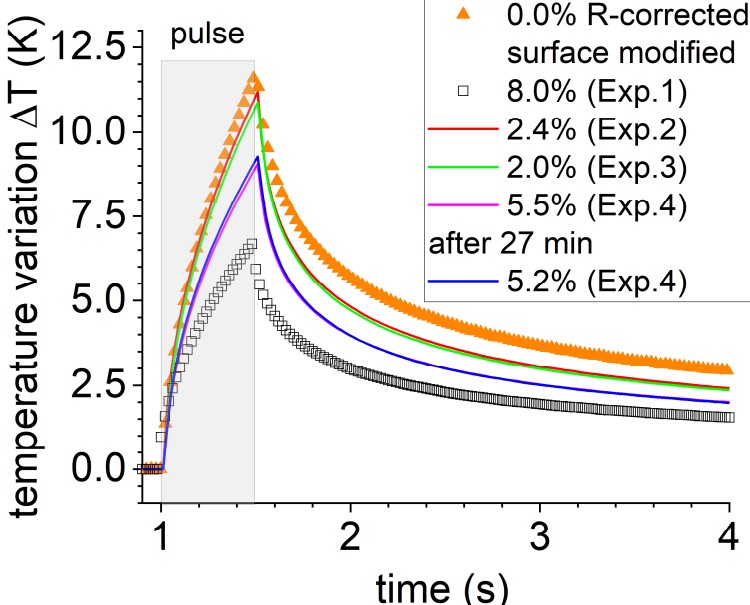

**Figure 17.** Surface temperature transients recorded in the type 1 sample (cut side) in different trials with different moisture contents. The 0% curve is shifted to higher values to correct for the increased reflection on the surface; the other curves are raw data without any correction.

Temperature transients from the formwork side of the type 1 and type 3 samples are displayed in Figures 19 and 20, respectively, showing the same effects in principle.

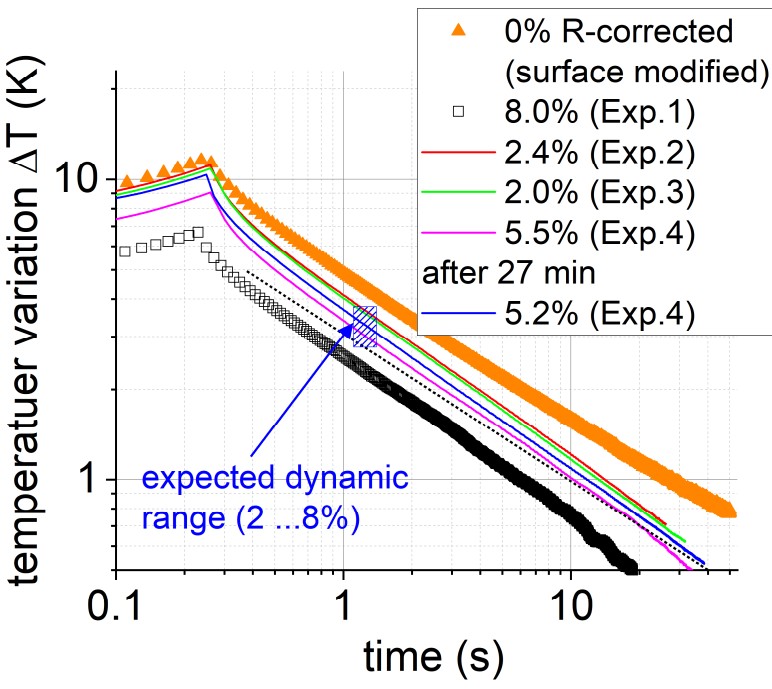

**Figure 18.** Same data as in Figure 17 as a log–log plot. The dotted line represents a curve according to Equation (1); the height of the blue striped rectangle indicates the expected moisture influence.

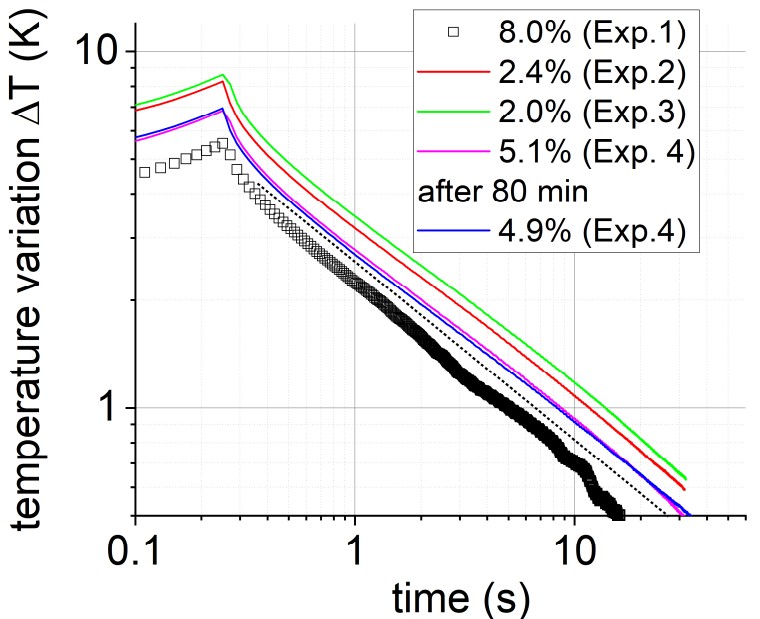

**Figure 19.** Surface temperature transients recorded in the type 1 sample (formwork side) in different trials with different moisture contents. The dotted line represents a curve according to Equation (1).

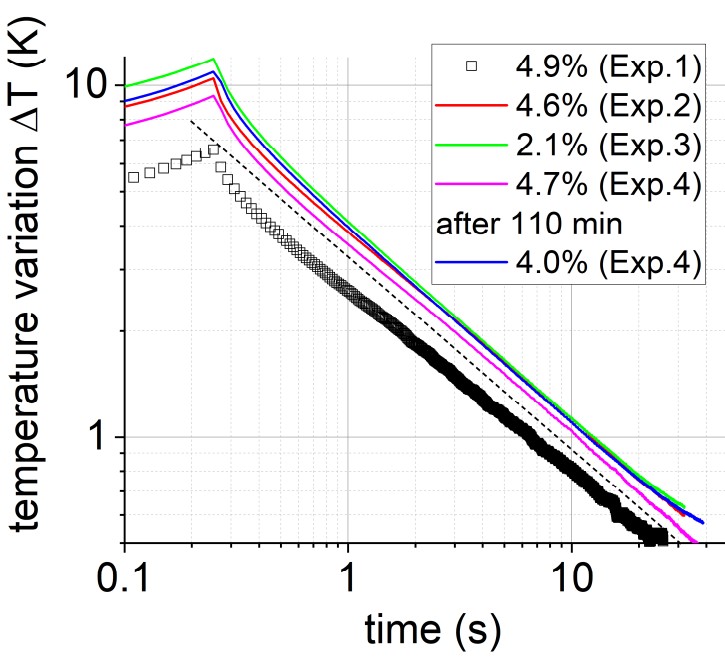

**Figure 20.** Surface temperature transients recorded in the type 3 sample in different trials with different moisture contents. The dotted line represents a curve according to Equation (1).

## 6. Discussion

The primary goal of the performed experiments was to investigate whether an expected change of the thermal effusivity of moist concrete is detectable by means of the photothermal method. The obtained results on moist concrete presented in Figures 17–20 demonstrate a clear influence of moisture on the respective surface temperature transient after pulsed heating. The size of the effect was even larger than estimated (considering the vertical extent of the blue rectangle in Figure 19) by a simplified model (Equation (1)), which worked well in the case of wet concrete. However, the estimated size of the effect is based on the listed assumptions about the influence of moisture on physical properties of concrete, mainly the linear relationship between moisture mass concentration and thermal conductivity with a slope value of 4. Selecting an higher value of 8, as reported for concrete in [25], would double the effect. It should be mentioned here that higher values of slope above 5 also have a large impact on the thermal diffusivity of moist concrete and could explain an increasing thermal diffusivity with increasing moisture as reported by Poblete [6].

Two further points have to be considered when the quantitative effect is discussed: the variable reflectivity and an (probably) inhomogeneous moisture and temperature distribution in depth.

- Since the exact reflectivity values were not measured for every photothermal measurement, Figures 17–20 show only raw data, i.e., measured temperature transients. The (additionally) observed moisture influence on the surface reflectivity (see Figure 16) would cause an additional effect when estimating the effusivity using Equation (1). Here, moist concrete tends to have a lower surface reflectivity, resulting in increased energy absorption (see Table 1). This would further (slightly) increase the differences between the estimated effusivities.
- Due to the dynamics of the drying process, inhomogeneous depth profiles for moisture and temperature are expected. The moisture profile is a violation of the first simplification for Equation (1) (isotropic material properties), and a temperature profile would violate the second simplification, as pointed out before. Therefore, Equation (1) can probably not be applied to moist materials, and the real effusivity cannot be

calculated—in contrast to dry materials. This conclusion also fits the observation that an investigation of moist samples with the transient plane source (TPS) revealed completely incomprehensible results. Since the detection principle of TSP is also based on thermal diffusion processes at and below the surface in a homogeneous material with isotropic temperature distribution, the underlying quantitative analysis might fail in the case of drying moist surfaces.

To achieve the goal of a quantitative measurement of moisture by the photothermal method, the influence of a spatially (in depth) varying effusivity profile and of an inhomogeneous temperature distribution have to be validated, for example, by thermal FEM simulations. Depending on the results, Equation (1) could be modified with a correction term or a new evaluation method could be deduced. The use of thermal effusivity as indicator for moisture may not useful, and another "thermal response parameter" may be better-suited. The reduced surface temperature (as shown in Figure 15) would probably be included in an extended model description of the photothermal response. Eventually, a systematic study of the named points is urgently needed to evaluate the performance and limits of the photothermal method for on-site applications.

### 7. Conclusions

This contribution presents a photothermal measurement concept to determine the thermal effusivity of a solid material, which might be later integrated in an instrument suited for on-site applications. The general principle was briefly explained herein, as well as the experimental setup. In order to investigate the influence of moisture in concrete on the measurement results, two concrete samples were soaked and dried several times. The samples were photothermally inspected in different stages of drying and soaking. These investigations revealed a clearly detectable effect of moisture on the measurement results. Thus, it is worth dealing with the proposed method in the future. The answer to the question in the headline is: YES! However, due to the specific conditions on moist surfaces, the underlying simple theory is no longer valid, and the effusivity of moist concrete could not be determined. More investigations are needed to achieve the goal of an on-site measurement instrument to detect moisture reliably.

**Funding:** This research received no external funding.

**Institutional Review Board Statement:** Not applicable.

**Informed Consent Statement:** Not applicable.

**Data Availability Statement:** All data can be provided upon request. In the case of thermograms, only a proprietary data format is available, but MATLAB or hdf5 format can be provided.

**Acknowledgments:** The author thanks Stefan Zirker, also from the Bundesanstalt für Materialforschung, for his efforts in the investigation by means of TPS. Special thanks to Mathias Ziegler for discussion and Somsubhro Chaudhuri for proofreading the manuscript.

**Conflicts of Interest:** The author declares no conflict of interest.

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
