# Peer review of "Influence of Moisture in Concrete on the Photothermal Response—A New Approach for a Measurement Method?"

_applsci, doi:10.3390/app13052768_

Round 1

Reviewer 1 Report

Attached.

Reviewer 2 Report

1. “During and after heating the surface of a solid, a temperature gradient arises between 73

the surface and the bulk leading to a heat flow into the volume of the solid. This heat flow 74

can be described by the well-known heat equation.”

Authors are advised to specify the equation which they are referring to.

2. Q=Qext ·(1-R)

Individual components of the equation must be clearly defined.

3. Authors may consider the aspect of composite mixing of nanomaterials with concrete and then throw some light on the interesting properties in the introduction section by citing the following articles. Ceramics International, 47, 11, 15293-15306, 2021.; Langmuir, 37, 31, 9356–9370, 2021.

Author Response

Thank you very much for your careful review with valuable comments. I tried to consider all points. I am sure the revised manuscript has now reached an improved quality. Please see my reply to your comments in detail:

  1. “During and after heating the surface of a solid, a temperature gradient arises between the surface and the bulk leading to a heat flow into the volume of the solid. This heat flow can be described by the well-known heat equation.”Authors are advised to specify the equation which they are referring to.

Answer: it is my pronounced opinion that scientific papers should not be loaded with knowledge from textbooks. Often, one finds formulas about blackbody radiation in thermographic research papers. It makes the manuscripts longer but not better if new results should be presented.

In case of the heat equation, it would require not only the simple equation, but also the introduction of the thermal diffusivity and an explanation about the later appearing Q (which makes the differential equation in homogeneously but can also be included in the boundary conditions). Therefore, my intention was to avoid this extension leading away from the actual topics of the manuscript. However, I introduced the references again: “well-known heat equation provided by numerous textbooks (see for example [20, 21]).”

  1. Q=Qext ·(1-R) Individual components of the equation must be clearly defined.

Answer: Q was introduced before, but I named it again. Qext is now addressed with the included word “external”, R is explained with a new sentence (“Here, R is like an effective parameter and results from the integral of the spectral intensity distribution of the source weighted with the reflection spectrum of the surface”), the last explanation is significant because R is usually defined for waves, thank you again for this hint!

  1. Authors may consider the aspect of composite mixing of nanomaterials with concrete and then throw some light on the interesting properties in the introduction section by citing the following articles. Ceramics International, 47, 11, 15293-15306, 2021.; Langmuir, 37, 31, 9356–9370, 2021.

Answer: There are some approaches to mixing nanoparticles with concrete to achieve specific properties, but this is beyond of my scope. The recommended papers are focused on thin layers and specific surface properties while the photothermal method is sensitive to the near surface region (a few mm). However, the evaluation of the penetration depth of the method is still open.

  1. The reviewer claimed in his review report form that the cited references “can be improved”:

I have checked the references again and have not any that are not appropriate. All references are needed to provide a specific statement. Perhaps the list of standards can shorten (but all are related to the content) or ref. 12 to reduce the number of references. Perhaps, the reviewer meant Ref. 22 (as self-referencing).  But here Eq.1 was successfully applied to various materials (including concrete) to evaluate the released energy Q of a flash lamp. So, Q was unknown and the effusivity was known. Thus, it was a completely different problem, but the same physics. And it was shown that the simple approach worked well. The reference could be removed from the manuscript without “damage”, but it provides the statement that the approach is useful.

  1. English language and style minor spell required:

The manuscript was proof-read by a native speaker, and a large number of corrections were made. For the sake of readability, I have created a second version in which all changes are indicated which is supplemented in the PDF-document.

Reviewer 3 Report

modify language 

Author Response

  1. English language and style minor spell required:

The manuscript was proof-read by a native speaker, and a large number of corrections were made. For the sake of readability, I have created a second version in which all changes are indicated which is supplemented in the PDF-document.

Round 2

Reviewer 1 Report

The authors revised their manuscript and addressed most of my comments perfectly and the revised manuscript could be accepted for publication.